# CLUE: Calibrated Latent Guidance for Offline Reinforcement Learning

**Jinxin Liu**[*]
Zhejiang University
Westlake University
`liujinxin@westlake.edu.cn`

**Lipeng Zu**[*]
Westlake University
`zulp@mail.ustc.edu.cn`

**Li He**
Westlake University
`heli.copter@foxmail.com`

**Donglin Wang**[†]
Westlake University
`wangdonglin@westlake.edu.cn`

**Abstract:** Offline reinforcement learning (RL) aims to learn an optimal policy from pre-collected and labeled datasets, which eliminates the time-consuming data collection in online RL. However, offline RL still bears a large burden of specifying/handcrafting extrinsic rewards for each transition in the offline data. As a remedy for the labor-intensive labeling, we propose to endow offline RL tasks with a few expert data and utilize the limited expert data to drive intrinsic rewards, thus eliminating the need for extrinsic rewards. To achieve that, we introduce **C**alibrated **L**atent g**U**idanc**E** (CLUE), which utilizes a conditional variational auto-encoder to learn a latent space such that intrinsic rewards can be directly qualified over the latent space. CLUE's key idea is to align the intrinsic rewards consistent with the expert intention via enforcing the embeddings of expert data to a calibrated contextual representation. We instantiate the expert-driven intrinsic rewards in sparse-reward offline RL tasks, offline imitation learning (IL) tasks, and unsupervised offline RL tasks. Empirically, we find that CLUE can effectively improve the sparse-reward offline RL performance, outperform the state-of-the-art offline IL baselines, and discover diverse skills from static reward-free offline data.

**Keywords:** Offline Reinforcement Learning, Intrinsic Rewards, Learning Skills

## 1  Introduction

Recent advances in reinforcement learning (RL) have shown great success in decision-making domains ranging from robot manipulation [1, 2] to navigation [3, 4] and large-language models [5]. Generally, an RL agent receives two sources of supervisory signals associated with the learning progress: 1) environment transition dynamics and 2) task-specifying rewards, where 1) the transition dynamics coordinate the agent's behaviors toward the environment affordances and 2) the task-specifying rewards capture the designer's preferences over agent behaviors. However, the two supervised signals themselves also limit the applicability of RL methods, since in many tasks, especially in real-world domains, either collecting online environmental transitions or labeling complex task-specifying rewards is time-consuming and laborious.

To tackle the above challenges, two separate RL branches have been proposed: 1) *offline RL* [6], also known as batch RL, which promises to learn effective policies from previously-collected static datasets without further online interaction, and 2) *intrinsic rewards* [7], which aim to capture a rich form of task knowledge (such as long-term exploration or exploitation) that provides additional guidance on how an agent should behave. Aligning with the task-specifying rewards, such intrinsic

---

[*]Jinxin Liu and Lipeng Zu contribute equally to this work. [†]Corresponding author: Donglin Wang.

7th Conference on Robot Learning (CoRL 2023), Atlanta, USA.

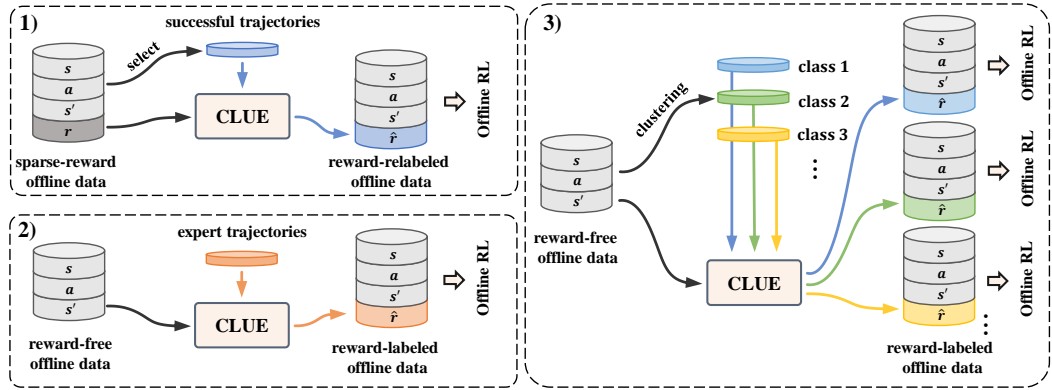

Figure 1: Three instantiations for the assumed "expert" data in offline RL settings: 1) sparse-reward offline RL, 2) offline imitation learning (IL) setting, and 3) unsupervised offline RL setting (aiming to learn diverse skills/policies from static reward-free offline data).

rewards promise to accelerate online RL by augmenting or replacing the manual task-specifying rewards (hereafter *extrinsic rewards*). In fact, prior offline RL methods [6] typically introduce a policy/value regularization and operate in a form of reward augmentation, which thus can be seen as a special kind of intrinsic motivation. However, such intrinsic motivation is only designed to eliminate the potential out-of-distribution (OOD) issues in offline RL and does not account for representing task-specifying behaviors (extrinsic rewards). In this work, we aim to design an offline RL intrinsic reward that promotes offline RL performance while representing task-specifying behaviors.

It is worth noting that adapting the online intrinsic rewards to offline RL problems is non-trivial. In online RL, intrinsic rewards often capture the long-term *temporal dependencies* of interaction trajectories [7, 8]. For example, Badia et al. [9] capture the novelty of states across multiple episodes; Eysenbach et al. [10] quantify the discriminability between skills represented by latent variables. However, such temporal dependencies rely on online interaction transitions, which thus cannot be straightly captured in offline settings. In this work, we thus propose to discard the above temporal dependence scheme and use an "expert" to facilitate labeling intrinsic rewards and guiding the offline agent. To do so, we identify three scenarios for the expert instantiations in offline RL settings:

**1)** For sparse-reward offline RL tasks, we filter out the trajectories that do not accomplish the tasks and take the success trajectories as the expert behaviors. By relabeling continuous intrinsic rewards for those failed trajectories, we expect such a reward relabelling procedure can promote offline learning.

**2)** For reward-free offline RL tasks, we assume that the agent has access to additional (limited) expert data generated by an expert policy. We expect that such limited expert data can provide useful intrinsic rewards for unlabeled transitions and then bias the learning policy toward expert behaviors.

**3)** Also considering the reward-free offline RL setting, we do not assume any additional expert data. Instead, we choose to cluster the offline transitions into a number of classes and take each class as a separate "expert". Then, we encourage offline agents to produce diverse behaviors when conditioned on different classes, in a similar spirit to the unsupervised skill learning in online RL.

We can see that in all three settings (Figure 1), we assume the existence of an expert (limited expert data), either from trajectory filtering, from an external expert, or through clustering. To instantiate the above intrinsic rewards, we propose **C**alibrated **L**atent g**U**idanc**E** (CLUE), which aims to label intrinsic rewards for unlabeled (or spare-reward) transitions in the offline RL setting. Specifically, CLUE uses a conditional variational auto-encoder to learn a latent space for both expert data and unlabeled data, then labels intrinsic rewards by computing the distance between the latent embeddings of expert and unlabeled data. CLUE's key idea is to explicitly bind together all the embeddings of expert data, thus learning a *calibrated* embedding for all expert behaviors. Intuitively, this binding procedure encourages the latent space to capture task-oriented behaviors such that latent space can produce task-oriented intrinsic guidance when computing distance over the latent space.

In summary, we make the following contributions in this paper: 1) We propose CLUE, which can provide pluggable intrinsic rewards for offline RL methods. 2) We demonstrate CLUE can effectively improve the spare-reward offline RL performance. 3) Considering offline imitation learning (IL) settings, CLUE can achieve better or comparable results compared to both the reward-labeled offline RL methods and the state-of-the-art offline IL methods. 4) We find that CLUE is able to discover diverse behaviors/skills in the unsupervised (reward-free) offline RL setting.

## 2 Related Works

The goal of our work is to learn task-oriented intrinsic rewards for sparse-reward or reward-free offline data. While there is a large body of research on learning rewards for RL tasks [11, 12, 13, 14, 15], most work assumes online RL settings, while we consider the offline RL setting. Additionally, little work has yet to verify intrinsic rewards across sparse-reward, IL, and unsupervised RL tasks together.

Typically, many intrinsic rewards have been proposed to encourage exploration in sparse-reward (online) RL tasks. In this case, intrinsic rewards are often formulated as state visitation counts [16, 17, 18], prediction error [19, 20], prediction uncertainty [21, 22], information gain [23], state entropy [24, 25, 26], and deviation from a default policy [27, 28]. However, these intrinsic rewards are often not well aligned with the task that the agent is solving. In contrast, the goal of our work is to learn a task-oriented intrinsic reward such that it promotes the policy learning progress for sparse-reward tasks.

Beyond the standard offline RL setup, learning from (static) reward-labeled offline data [6, 29, 30, 31, 32], offline imitation learning (IL) considers learning from expert trajectories and (reward-free) sub-optimal offline data, which can be generally folded into two paradigms [33]: behavior cloning (BC) and offline inverse RL (IRL). BC directly learns a policy from expert trajectories using supervised learning [34]. Due to compounding errors induced by covariate shift [35], BC methods require a large amount of expert data, thus hindering the application on data-scarce scenarios. To overcome such limitations, offline IRL methods consider matching the state-action distributions induced by the expert [36, 37, 38, 39, 40, 41]. Typically, they formulate the expert matching objective by introducing a discriminator and trying to find the saddle point of a min-max optimization, which tends to be brittle and sensitive to the training (offline) data. However, our CLUE does not introduce any adversarial objective, thus exhibiting more robust performance on a wide variety of tasks.

The idea of unsupervised RL is to learn diverse behaviors/skills in an open-ended environment without access to extrinsic rewards [42, 43, 44]. Previous unsupervised RL methods are often formulated through the lens of empowerment [45]. Central to this formulation is the information-theoretic skill discovery approach, where diverse skills can be discovered by optimizing the long-term temporal dependencies of interaction trajectories, *e.g.*, maximizing the mutual information between induced trajectories and some latent/context variables [10, 46, 47, 48, 49, 50, 51]. In this work, we propose to discard this online temporal dependence scheme and use clustering methods to formulate such diversity and use CLUE to label intrinsic rewards to guide offline agents.

## 3 Preliminary

**Offline RL.** We consider RL in a Markov Decision Process (MDP) $\mathcal{M} := (\mathcal{S}, \mathcal{A}, T, r, p_0, \gamma)$, where $\mathcal{S}$ is the state space, $\mathcal{A}$ is the action space, $T$ is the environment transition dynamics, $r$ is the task-oriented extrinsic reward function, $p_0$ is the initial state distribution, and $\gamma$ is the discount factor. The goal of RL is to find an optimal policy $\pi_\theta(\mathbf{a}|\mathbf{s})$ that maximizes the expected return $\mathbb{E}_{\pi_\theta(\tau)} \left[ \sum_{t=0}^{\infty} \gamma^t r_t \right]$ when interacting with the environment $\mathcal{M}$, where trajectory $\tau := (\mathbf{s}_0, \mathbf{a}_0, r_0, \mathbf{s}_1, \cdots)$ denotes the generated trajectory, $\mathbf{s}_0 \sim p_0(\mathbf{s}_0)$, $\mathbf{a}_t \sim \pi_\theta(\mathbf{a}_t|\mathbf{s}_t)$, $\mathbf{s}_{t+1} \sim T(\mathbf{s}_{t+1}|\mathbf{s}_t, \mathbf{a}_t)$, and $r_t$ denotes the extrinsic reward $r(\mathbf{s}_t, \mathbf{a}_t)$ at time step $t$. In offline RL, the agent can not interact with the environment and only receives a static dataset of trajectories $\mathcal{D} := \{\tau_i\}_i^n$, pre-collected by one or a mixture of (unknown) behavior policies. Then, the goal of offline RL is to find the best policy from offline data.

**Conditional variational auto-encoders (CVAE).** Given offline data $\mathbf{x}$, the variational auto-encoder (VAE) [52] proposes to maximize the variational lower bound,

$$\log p_\theta(\mathbf{x}) = \text{KL}(q_\phi(\mathbf{z}|\mathbf{x})\|p_\theta(\mathbf{z}|\mathbf{x})) + \mathbb{E}_{q_\phi(\mathbf{z}|\mathbf{x})}\left[-\log q_\phi(\mathbf{z}|\mathbf{x}) + \log p_\theta(\mathbf{x},\mathbf{z})\right] \tag{1}$$

$$\geq -\text{KL}(q_\phi(\mathbf{z}|\mathbf{x})\|p_\theta(\mathbf{z})) + \mathbb{E}_{q_\phi(\mathbf{z}|\mathbf{x})}\left[\log p_\theta(\mathbf{x}|\mathbf{z})\right], \tag{2}$$

where $p_\theta(\mathbf{z})$ is the prior distribution, $q_\phi(\mathbf{z}|\mathbf{x})$ denotes the encoder model, and $p_\theta(\mathbf{x}|\mathbf{z})$ denotes decoder model. Considering the structured output prediction settings, conditional VAE (CVAE) maximizes the variational lower bound of the conditional log-likelihood:

$$\log p_\theta(\mathbf{x}|\mathbf{y}) \geq -\text{KL}(q_\phi(\mathbf{z}|\mathbf{x},\mathbf{y})\|p_\theta(\mathbf{z}|\mathbf{y})) + \mathbb{E}_{q_\phi(\mathbf{z}|\mathbf{x},\mathbf{y})}\left[\log p_\theta(\mathbf{x}|\mathbf{z},\mathbf{y})\right]. \tag{3}$$

## 4 CLUE: Calibrated Latent Guidance

In this section, we introduce our method CLUE (**C**alibrated **L**atent g**U**idanc**E**) that learns a calibrated latent space such that intrinsic rewards can be directly gauged over the latent space. We begin by assuming access to limited expert offline data and describe how we can use it to label intrinsic rewards for reward-free offline data in Section 4.1. Next in Section 4.2, we describe three offline RL instantiations, including one sparse-reward and two reward-free (offline imitation learning and unsupervised offline RL) settings, each corresponding to a scenario discussed previously (Figure 1).

### 4.1 Calibrated Intrinsic Rewards

Assuming access to limited expert offline data $\mathcal{D}^e := \{(\mathbf{s}, \mathbf{a}, \mathbf{s}')\}$ and a large number of reward-free offline data $\mathcal{D} := \{(\mathbf{s}, \mathbf{a}, \mathbf{s}')\}$, our goal is to use $\mathcal{D}^e$ to learn an intrinsic reward function $\hat{r}(\mathbf{s}, \mathbf{a})$ for the reward-free transitions in $\mathcal{D}$, such that we can recover expert behaviors using the relabeled offline data $\mathcal{D}_{\hat{r}} := \{(\mathbf{s}, \mathbf{a}, \hat{r}, \mathbf{s}')\}$. With a slight abuse of notation, here we write $\hat{r}$ in transitions $\{(\mathbf{s}, \mathbf{a}, \hat{r}, \mathbf{s}')\}$ to denote the relabeled intrinsic reward $\hat{r}(\mathbf{s}, \mathbf{a})$.

We first use CVAE to model the mixed offline behaviors in $\mathcal{D}^e \cup \mathcal{D}$. Specifically, we take state $\mathbf{s}$ as the input/conditional variable and take action $\mathbf{a}$ as the prediction variable. For each behavior samples $(\mathbf{s}, \mathbf{a})$ in mixed data $\mathcal{D}^e \cup \mathcal{D}$, we maximize the following variational lower bound:

$$\log p_\theta(\mathbf{a}|\mathbf{s}) \geq -\text{KL}(q_\phi(\mathbf{z}|\mathbf{s},\mathbf{a})\|p_\theta(\mathbf{z}|\mathbf{s})) + \mathbb{E}_{q_\phi(\mathbf{z}|\mathbf{a},\mathbf{s})}\left[\log p_\theta(\mathbf{a}|\mathbf{z},\mathbf{s})\right] \tag{4}$$

$$\approx -\text{KL}(q_\phi(\mathbf{z}|\mathbf{s},\mathbf{a})\|p_\theta(\mathbf{z}|\mathbf{s})) + \frac{1}{L}\sum_{l=1}^{L}\log p_\theta(\mathbf{a}|\mathbf{z}^{(l)},\mathbf{s}) \triangleq \mathcal{L}_{\text{CVAE}}(\mathbf{s},\mathbf{a};\theta,\phi), \tag{5}$$

where $\mathbf{z}^{(l)} \sim \mathcal{N}(\mathbf{z}|\mu_\phi(\mathbf{s},\mathbf{a}), \sigma_\phi^2(\mathbf{s},\mathbf{a}))$ [2], $L$ is the number of samples, and $\mathcal{L}_{\text{CVAE}}(\mathbf{s},\mathbf{a};\theta,\phi)$ is the corresponding empirical lower bound. For simplicity, we set the prior distribution as the standard Gaussian distribution, *i.e.*, $p_\theta(\mathbf{z}|\mathbf{s}) = \mathcal{N}(\mathbf{0},\mathbf{1})$.

For a query sample $(\mathbf{s}, \mathbf{a})$, we label its intrinsic reward $\hat{r}(\mathbf{s}, \mathbf{a})$ by computing the negative distance between the latent embeddings of the expert data and the query sample,

$$\hat{r}(\mathbf{s},\mathbf{a}) = \exp\left(-c \cdot \|\mathbf{z}_e - \mathbf{z}(\mathbf{s},\mathbf{a})\|^2\right), \tag{6}$$

where $\mathbf{z}_e = \mathbb{E}_{(\mathbf{s},\mathbf{a})\sim\mathcal{D}^e}\left[q_\phi(\mathbf{z}|\mathbf{s},\mathbf{a})\right]$, $\mathbf{z}(\mathbf{s},\mathbf{a}) \sim q_\phi(\mathbf{z}|\mathbf{s},\mathbf{a})$, and $c > 0$ is a temperature factor.

However, naively maximizing $\mathcal{L}_{\text{CVAE}}$ in Equation 5 may lead to undesirable embeddings with

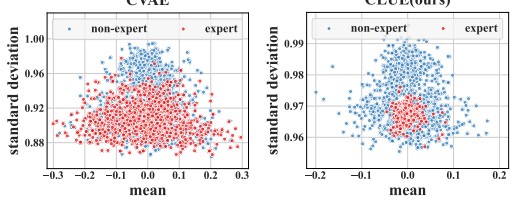

Figure 2: Latent embeddings of expert and non-expert offline data on D4RL *antmaze-medium-diverse-v2* dataset, where embeddings are learned by the naive CVAE (*left*) and our CLUE (*right*).

varying scales that do not capture task-relevant behaviors when computing the latent distance. For example, in Figure 2 *left*, we visualize the embeddings of the expert data $\mathcal{D}^e$ and the unlabeled (non-expert) offline data $\mathcal{D}$. We can see that the embeddings for the expert and non-expert data are generally mixed together without clear separation. There is a large variance in expert data embeddings,

---

[2]We define $\mu_\phi(\mathbf{s},\mathbf{a})$ and $\sigma_\phi(\mathbf{s},\mathbf{a})$ to be feed-forward networks with parameters $\phi$, taking concatenated $\mathbf{s}$ and $\mathbf{a}$, and outputting the parameters (mean and std) of a Gaussian distribution in the latent space, respectively.

and directly estimating the mean of the expert embeddings (*i.e.*, $\mathbf{z}_e = \mathbb{E}_{(\mathbf{s},\mathbf{a})\sim\mathcal{D}^e}[q_\phi(\mathbf{z}|\mathbf{s},\mathbf{a})]$) cannot effectively represent task-oriented behaviors, causing the labeled intrinsic reward $\hat{r}(\mathbf{s},\mathbf{a})$ to be biased.

To guarantee the intrinsic reward formulation $\hat{r}(\mathbf{s},\mathbf{a})$ in Equation 6 to be task-oriented, we thus propose to learn calibrated embeddings. To do so, we explicitly bind together the expert embeddings, expecting the expert embeddings to "collapse" into a single embedding. Thus, we introduce the following calibration regularization over expert embeddings:

$$\min \mathcal{L}_{\text{calibr}} := \mathbb{E}_{(\mathbf{s},\mathbf{a})\sim\mathcal{D}^e}\left[\|\mu_\phi(\mathbf{s},\mathbf{a})\|^2 + \|\sigma_\phi(\mathbf{s},\mathbf{a})\|^2\right]. \tag{7}$$

Due to the standard Gaussian prior for $p_\theta(\mathbf{z}|\mathbf{s})$ in Equation 5, we constrain not only the variance of the expert embeddings but also the mean of the expert in Equation 7. Intuitively, $\mathcal{L}_{\text{calibr}}$ unifies expert embeddings ("collapsed" to a single point), therefore providing effective $\mathbf{z}_e$ when computing intrinsic rewards $\hat{r}(\mathbf{s},\mathbf{a})$ in Equation 6. As shown in Figure 2 *right*, the expert embeddings and their mean are almost bound to a single point, so we can directly measure intrinsic rewards in latent space.

## 4.2 Intrinsic Reward Instantiations

Here we describe three offline instantiations, one spare-reward, and two reward-free offline settings, that permit us to meet the previous expert data assumption (in Section 4.1) and label intrinsic rewards.

**Spare-reward offline RL.** Considering the challenging spare-reward offline data $\{(\mathbf{s},\mathbf{a},r,\mathbf{s}')\}$, we can filter out the unsuccessful trajectories and take the finished trajectories as the expert data. Then, we can replace the original spare rewards with the learned continuous intrinsic rewards.

**Offline IL.** Considering the reward-free offline RL data $\{(\mathbf{s},\mathbf{a},\mathbf{s}')\}$, we can assume the agent has access to additional expert data (as few as only one trajectory). Then we can use the learned intrinsic rewards to relabel the reward-free transitions, obtaining labeled offline data $\{(\mathbf{s},\mathbf{a},\hat{r},\mathbf{s}')\}$.

**Unsupervised offline RL.** Given the reward-free offline data $\{(\mathbf{s},\mathbf{a},\mathbf{s}')\}$, we can use clustering algorithms to cluster the data into multiple classes and then treat each class as *separate "expert" data*. In this way, we expect to learn different skills/policies when conditioned in different classes.

## 5 Experiments

In this section, we first empirically demonstrate the advantages of our pluggable intrinsic rewards in sparse-reward offline RL tasks. Second, we then evaluate our CLUE in offline IL tasks, studying how effective our intrinsic reward is in contrast to a broad range of state-of-the-art offline imitation learning methods. Third, we study CLUE in unsupervised offline RL settings, expecting to discover diverse behaviors from static offline data. Finally, we conduct ablation studies on the calibration regularization and the amount of unlabeled offline data. All of our results are evaluated over 10 random seeds and 10 episodes for each seed.

**Implementation.** Note that our intrinsic rewards are pluggable and can be combined with any offline RL algorithms. In our implementation, we combine CLUE with the Implicit Q-Learning (IQL) algorithm [53] which is one of the state-of-the-art offline algorithms and can solve most of the (reward-labeled) offline tasks with competitive performance. Our base IQL implementation is adapted from IQL[3], and we set all hyperparameters to the ones recommended in the original IQL paper.

### 5.1 Sparse-Reward Offline RL Tasks

Here we evaluate CLUE on spares-reward tasks (including AntMaze and Adroit domains) from the D4RL benchmark. In this setting, we use the inherent sparse rewards in the dataset to select expert data, *i.e.*, selecting the task-completed or goal-reached trajectories from the sparse-reward dataset and treating them as the expert data. In Tables 1 and 2, we compare our method with baseline methods (IQL [53] and OTR [54]) when only one expert trajectory[4] is selected. For comparison, we train

---

[3]https://github.com/ikostrikov/implicit_q_learning.

[4]In the appendix, we also compare our method with baseline methods when at most 10 completed trajectories are selected, where "at most 10" refers to that there may be less than 10 successful trajectories in D4RL dataset.

Table 1: Normalized scores (mean and standard deviation) of CLUE and baselines on sparse-reward AntMaze tasks, where both OTR and CLUE use IQL as the base offline RL algorithm. The highest scores between our CLUE and baseline OTR are highlighted.

| Dataset | IQL | OTR | CLUE |
|---|---|---|---|
| umaze | 88.7 | 81.6 ± 7.3 | 92.1 ± 3.9 |
| umaze-diverse | 67.5 | 70.4 ± 8.9 | 68.0 ± 11.2 |
| medium-play | 72.9 | 73.9 ± 6.0 | 75.3 ± 6.3 |
| medium-diverse | 72.1 | 72.5 ± 6.9 | 74.6 ± 7.5 |
| large-play | 43.2 | 49.7 ± 6.9 | 55.8 ± 7.7 |
| large-diverse | 46.9 | 48.1 ± 7.9 | 49.9 ± 6.9 |
| AntMaze-v2 total | 391.3 | 396.2 | **415.7** |

Table 2: Normalized scores (mean and standard deviation) of CLUE and baselines on sparse-reward Adroit tasks, where the highest scores between CLUE and OTR are highlighted.

| Dataset | IQL | OTR | CLUE |
|---|---|---|---|
| door-cloned | 1.6 | 0.01 ± 0.01 | 0.02 ± 0.01 |
| door-human | 4.3 | 5.9 ± 2.7 | 7.7 ± 3.9 |
| hammer-cloned | 2.1 | 0.9 ± 0.3 | 1.4 ± 1.0 |
| hammer-human | 1.4 | 1.8 ± 1.4 | 1.9 ± 1.2 |
| pen-cloned | 37.3 | 46.9 ± 20.9 | 59.4 ± 21.1 |
| pen-human | 71.5 | 66.8 ± 21.2 | 82.9 ± 20.2 |
| relocate-cloned | -0.2 | -0.24 ± 0.03 | -0.23 ± 0.02 |
| relocate-human | 0.1 | 0.1 ± 0.1 | 0.2 ± 0.3 |
| Adroit-v0 total | 118.1 | 122.2 | **153.3** |

Table 3: Normalized scores (mean and standard deviation) of CLUE and baselines on locomotion tasks using one (K=1), five (K=5), and ten (K=10) expert demonstrations. Both CLUE and OTR uses IQL as the base offline RL algorithm, and we highlight the highest score in each setting.

| Dataset | IQL | OTR (K=1) | CLUE (K=1) | OTR (K=5) | CLUE (K=5) | OTR (K=10) | CLUE (K=10) |
|---|---|---|---|---|---|---|---|
| halfcheetah-medium | 47.4 ± 0.2 | 43.3 ± 0.2 | 45.6 ± 0.3 | 43.3 ± 0.2 | 45.2 ± 0.2 | 43.1 ± 0.3 | 45.7 ± 0.2 |
| halfcheetah-medium-replay | 44.2 ± 1.2 | 41.3 ± 0.6 | 43.5 ± 0.5 | 41.9 ± 0.3 | 43.2 ± 0.4 | 41.6 ± 0.3 | 43.2 ± 0.5 |
| halfcheetah-medium-expert | 86.7 ± 5.3 | 89.6 ± 3.0 | 90.0 ± 2.4 | 89.9 ± 1.9 | 91.9 ± 1.4 | 87.9 ± 3.4 | 91.0 ± 2.5 |
| hopper-medium | 66.2 ± 5.7 | 78.7 ± 5.5 | 78.3 ± 5.4 | 79.5 ± 5.3 | 79.1 ± 3.5 | 80.0 ± 5.2 | 79.9 ± 6.0 |
| hopper-medium-replay | 94.7 ± 8.6 | 84.8 ± 2.6 | 94.3 ± 6.0 | 85.4 ± 1.7 | 93.3 ± 4.5 | 84.4 ± 1.8 | 93.7 ± 4.1 |
| hopper-medium-expert | 91.5 ± 14.3 | 93.2 ± 20.6 | 96.5 ± 14.7 | 90.4 ± 21.5 | 104.0 ± 5.4 | 96.6 ± 21.5 | 102.3 ± 7.7 |
| walker2d-medium | 78.3 ± 8.7 | 79.4 ± 1.4 | 80.7 ± 1.5 | 79.8 ± 1.4 | 79.6 ± 0.7 | 79.2 ± 1.3 | 81.7 ± 1.2 |
| walker2d-medium-replay | 73.8 ± 7.1 | 66.0 ± 6.7 | 76.3 ± 2.8 | 71.0 ± 5.0 | 75.1 ± 1.3 | 71.8 ± 3.8 | 75.3 ± 4.6 |
| walker2d-medium-expert | 109.6 ± 1.0 | 109.3 ± 0.8 | 109.3 ± 2.1 | 109.4 ± 0.4 | 109.9 ± 0.3 | 109.6 ± 0.5 | 110.7 ± 0.2 |
| locomotion-v2 total | 692.4 | 685.6 | **714.5** | 690.6 | **721.3** | 694.2 | **723.5** |

IQL over the naive sparse-reward D4RL data and train OTR over the relabeled D4RL dataset (using optimal transport to compute intrinsic rewards and employing IQL to learn offline RL policy). We can find that in 13 out of 14 tasks across AntMaze and Adroit domains, our CLUE outperforms the baseline OTR. Meanwhile, compared to naive IQL (with sparse rewards), our CLUE implementation obtains a total score of 106.2% on AntMaze tasks and 129.8% on Adroit tasks. This means that with only a single expert trajectory, we can completely replace the *sparse rewards* with our intrinsic reward in offline RL tasks, which can even achieve higher performance.

## 5.2   Offline Imitation Learning Tasks

Then, we evaluate CLUE on offline IL tasks. We continue to use the D4RL data as offline data, but here we explicitly discard the reward signal. Then, we use SAC to train an online expert policy to collect expert demonstrations in each environment. We first compare CLUE to 1) naive IQL with the (ground-truth) reward-labeled offline data and 2) OTR under our offline IL setting. In Table 3, we provide the comparison results with 1, 5, and 10 expert trajectories. We can see that in 22 out of 27 offline IL settings, CLUE outperforms (or performs equally well) the most related baseline OTR, demonstrating that CLUE can produce effective intrinsic rewards. Meanwhile, with only a single expert trajectory, our CLUE implementation can achieve 103.2% of the total scores of naive IQL in locomotion tasks. This means that with only one expert trajectory, our intrinsic rewards can even replace the *continuous* ground-truth rewards in offline RL tasks and enable better performance.

Next, we compare CLUE to a representative set of offline IL baselines: SQIL[55] with TD3+BC [56] implementation, ORIL [57], IQ-Learn [58], ValueDICE [37], DemoDICE [59], and SMODICE [60]. Note that the original SQIL is an online IL method, here we replace its (online) base RL algorithm with TD3+BC, thus making it applicable to offline tasks. In Table 4, we provide the comparison results over D4RL locomotion tasks. We can see that, overall, our CLUE performs better than most offline IL baselines, showing that our intrinsic reward is well capable of capturing expert behaviors. Further, we point out that CLUE is also robust to offline data with different qualities: most previous

Table 4: Normalized scores (mean and standard deviation) of CLUE and offline IL baselines on MuJoCo locomotion tasks using one expert trajectory (K=1) and ten expert trajectories (K=10). We highlight the scores that are within two points of the highest score.

| | Dataset | SQIL | IQ-Learn | ORIL | ValueDICE | DemoDICE | SMODICE | **CLUE** |
|---|---|---|---|---|---|---|---|---|
| **K=1** | halfcheetah-medium | 24.3 ± 2.7 | 21.7 ± 1.5 | 56.8 ± 1.2 | 36.4 ± 1.7 | 42.0 ± 0.8 | 42.4 ± 0.6 | 45.6 ± 0.3 |
| | halfcheetah-medium-replay | 43.9 ± 1.0 | 7.7 ± 1.6 | 46.2 ± 1.1 | 29.4 ± 3.0 | 38.3 ± 1.3 | 38.3 ± 2.0 | 43.5 ± 0.5 |
| | halfcheetah-medium-expert | 6.7 ± 1.2 | 2.0 ± 0.4 | 48.7 ± 2.4 | 1.0 ± 2.4 | 66.2 ± 4.3 | 80.9 ± 2.3 | 90.0 ± 2.4 |
| | hopper-medium | 66.9 ± 5.1 | 29.6 ± 5.2 | 96.3 ± 0.9 | 44.0 ± 12.3 | 56.4 ± 1.9 | 54.8 ± 1.2 | 78.3 ± 5.4 |
| | hopper-medium-replay | 98.6 ± 0.7 | 23.0 ± 9.4 | 56.7 ± 12.9 | 52.5 ± 14.4 | 70.7 ± 8.5 | 30.4 ± 7.8 | 94.3 ± 6.0 |
| | hopper-medium-expert | 13.6 ± 9.6 | 9.1 ± 2.2 | 25.1 ± 12.8 | 27.3 ± 10.0 | 103.7 ± 5.5 | 82.4 ± 7.7 | 96.5 ± 14.7 |
| | walker2d-medium | 51.9 ± 11.7 | 5.7 ± 4.0 | 20.4 ± 13.6 | 13.9 ± 9.1 | 74.5 ± 2.6 | 67.8 ± 6.0 | 80.7 ± 1.5 |
| | walker2d-medium-replay | 42.3 ± 5.8 | 17.0 ± 7.6 | 71.8 ± 9.6 | 52.7 ± 13.1 | 57.2 ± 8.7 | 49.7 ± 4.6 | 76.3 ± 2.8 |
| | walker2d-medium-expert | 18.8 ± 13.1 | 7.7 ± 2.4 | 11.6 ± 14.7 | 37.3 ± 13.7 | 87.3 ± 10.5 | 94.8 ± 11.1 | 109.3 ± 2.1 |
| **K=10** | halfcheetah-medium | 48.0 ± 0.3 | 29.2 ± 6.4 | 56.7 ± 0.9 | 40.0 ± 1.9 | 41.9 ± 0.5 | 41.6 ± 0.7 | 45.7 ± 0.2 |
| | halfcheetah-medium-replay | 45.1 ± 0.5 | 29.6 ± 3.1 | 46.2 ± 0.6 | 39.6 ± 1.0 | 38.5 ± 1.6 | 39.3 ± 0.9 | 43.2 ± 0.5 |
| | halfcheetah-medium-expert | 11.4 ± 4.7 | 2.9 ± 0.8 | 46.6 ± 6.0 | 25.2 ± 8.3 | 67.1 ± 5.5 | 89.4 ± 1.4 | 91.0 ± 2.5 |
| | hopper-medium | 65.8 ± 4.1 | 31.6 ± 6.2 | 101.5 ± 0.6 | 37.6 ± 9.5 | 57.4 ± 1.7 | 55.9 ± 1.7 | 79.9 ± 6.0 |
| | hopper-medium-replay | 96.6 ± 0.7 | 38.0 ± 7.0 | 29.0 ± 6.8 | 83.6 ± 8.9 | 56.9 ± 4.6 | 32.6 ± 8.7 | 93.7 ± 4.1 |
| | hopper-medium-expert | 19.6 ± 9.4 | 19.3 ± 3.9 | 18.9 ± 9.0 | 28.4 ± 8.6 | 96.9 ± 8.5 | 89.3 ± 5.9 | 102.3 ± 7.7 |
| | walker2d-medium | 72.4 ± 8.6 | 46.4 ± 8.5 | 82.3 ± 8.8 | 54.3 ± 8.6 | 71.3 ± 4.3 | 67.9 ± 7.9 | 81.7 ± 1.2 |
| | walker2d-medium-replay | 82.4 ± 4.7 | 16.6 ± 9.3 | 70.0 ± 10.0 | 54.6 ± 9.6 | 58.1 ± 7.9 | 52.4 ± 6.8 | 75.3 ± 4.6 |
| | walker2d-medium-expert | 12.5 ± 9.3 | 24.5 ± 4.8 | 6.5 ± 8.2 | 40.1 ± 9.4 | 103.5 ± 7.8 | 107.5 ± 1.0 | 110.7 ± 0.2 |

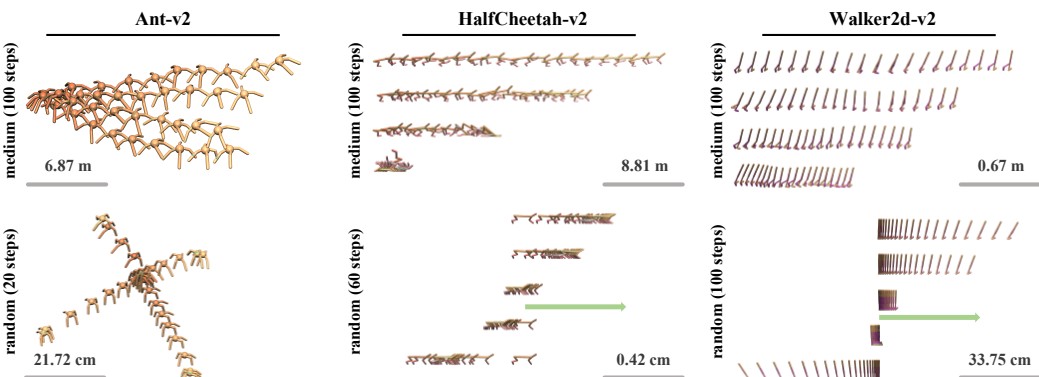

Figure 3: Qualitative visualizations of the learned skills in Ant, HalfCheetah, and Walker domains. We can see that the ant learns to move in different directions, the half-cheetah learns to flip upright and run at different speeds, and the walker learns to walk at different speeds.

adversarial-based methods deliberately depict unlabeled offline data as sub-optimal and tag expert data as optimal, which can easily lead to a biased policy/discriminator. For example, we can see that ORIL's performance deteriorates severely on all medium-expert tasks. On the contrary, CLUE does not bring in any adversarial objectives and is therefore much more robust.

## 5.3 Unsupervised Offline RL Tasks

Considering the reward-free offline RL settings, here we expect to learn diverse skills/policies from the static offline data. To do so, we first use K-means clustering to cluster similar transitions and treat the transitions in each of the clustered classes as separate expert data. For each class, we then use CLUE to learn the corresponding intrinsic reward function and label intrinsic rewards for the rest of the unlabelled data to learn corresponding skills. We visualize the learned diverse behaviors in Figure 3 (see videos in the supplementary material). We can see that our CLUE+K-means implementation provides a promising solution to unsupervised offline RL, which successfully produces a diverse set of skills and thus illustrates a huge potential for skill discovery from static offline data.

## 5.4 Ablation Studies

**Ablating the calibration regularization.** The key idea of CLUE is to encourage learning calibrated expert embeddings, thus providing effective embedding space when computing intrinsic rewards.

Here we verify this intuition by ablating the calibration regularization $\mathcal{L}_{\textbf{calibr}}$ in Equation 7, and directly using CVAE to learn the embedding space. We show ablation results in Figure 4. We can observe that the naive CVAE implementation (ablating the calibration regularization $\mathcal{L}_{\textbf{calibr}}$) suffers from a significant performance drop compared with our CLUE, indicating that our calibration regularization is effective in promoting embedding alignment and producing task-oriented intrinsic rewards.

**Varying the amount of unlabeled offline data.**
To assess the effectiveness of our intrinsic rewards under data-scarce scenarios, here we vary the amount of unlabeled offline data available for offline IL settings (see results for sparse-reward settings in the appendix). In Figure 5, we show the normalized results with a small number of D4RL data ranging from 5% ∼ 25%. We can see that across a range of dataset sizes, CLUE can perform well and achieve competitive performance compared to the state-of-the-art offline IL baselines.

Figure 4: Ablating the effect of the calibration regularization. We can see that ablating the regularization generally causes performance degradation, implying that our calibration regularization can indeed encourage task-oriented intrinsic rewards. u: umaze. m: medium. l: large. d: diverse. p: play.

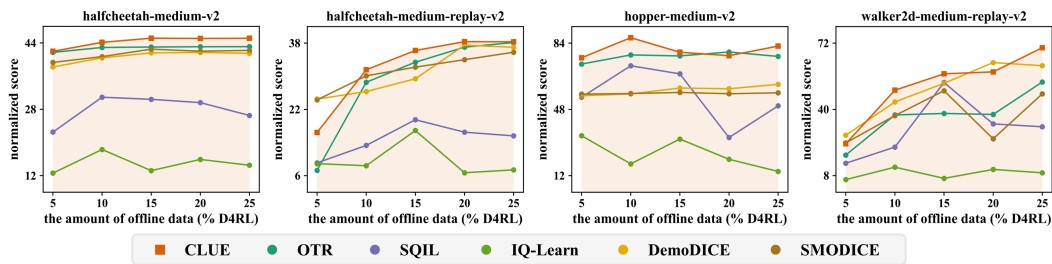

Figure 5: Ablating the number of unlabeled offline data. To compare CLUE with the baseline methods, we shade the area below the scores of our CLUE. We can see that CLUE's performance generally outperforms offline IL baselines across a range of domains and dataset sizes in each domain.

## 6 Discussion and Limitations

In this paper, we propose calibrated latent guidance (CLUE) algorithm which labels intrinsic rewards for unlabeled offline data. CLUE is an effective method and can provide pluggable intrinsic rewards compatibility with any offline RL algorithms that require reward-annotated data for offline learning. We have demonstrated that CLUE can effectively improve the spare-reward offline RL performance, achieve competitive performance compared with the state-of-the-art baselines in offline IL tasks, and learn diverse skills in unsupervised offline RL settings.

**Future work and limitations.** Our CLUE formulation assumes the presence of a large batch of offline data and expert trajectories. This setting is common in many robotic domains. However, we also point out that in some tasks, expert trajectories may be state-only and not contain actions. Also, there may be transition dynamics shifts between the expert data and the unlabelled offline data in some cross-domain problem settings. Thus, future directions could investigate the state-only expert data and the cross-domain intrinsic rewards. In view of the fact that our intrinsic rewards are measured in the latent space, it is feasible to apply our CLUE approach in both the state-only and cross-domain scenarios, as long as we impose the corresponding regularization on the learned latent embeddings. For example, we can directly add cross-domain constraints [61, 62] over our calibration regularization, making CLUE suitable for cross-domain tasks. In summary, we believe future work to this work will contribute to a general reward-relabeling framework capable of labeling effective intrinsic rewards and addressing more realistic robotic tasks *e.g.*, discovering diverse robot behaviors from static manipulation data and transferring offline cross-domain behaviors in sim2real tasks.

**Acknowledgments**

This work was supported by the National Science and Technology Innovation 2030 - Major Project (Grant No. 2022ZD0208800), and NSFC General Program (Grant No. 62176215).

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

# 7 Additional Results

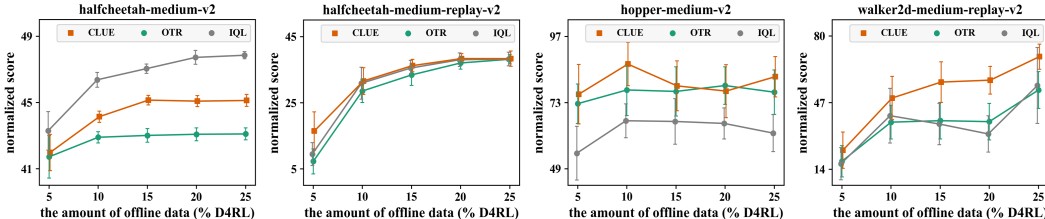

Figure 6: Ablating the number of unlabeled trajectories. We investigate the effect of unlabeled trajectories on the performance. CLUE's performance generally outperforms OTR. Further, we can see that CLUE approximates the vanilla IQL method (with D4RL rewards) more closely and can even outperform IQL given such a lack of offline data ($\leq 25\%$).

**Varying the amount of unlabeled offline data.** Here we vary the amount of unlabeled offline data available for sparse-reward settings. Figure 6 shows that adding more unlabeled data improves the performance of both CLUE and OTR. However, across a range of offline imitation tasks, CLUE shows better performance compared to OTR. We also plot the performance curve of naive IQL with (reward-labeled) offline data in Figure 6. We can see that with extremely limited offline data ($\leq 25\%$), CLUE approaches IQL's performance more closely on the halfcheetah-medium task, and can even outperform IQL on the remaining three tasks.

Table 5: Using 10% of D4RL data, normalized scores (mean and standard deviation) of CLUE and baselines on antmaze tasks using one (K=1) and ten (K=10) expert demonstrations. The expert trajectories are picked from the chosen 10% dataset. The highest score in each setting is highlighted.

| Dataset | IQL | OTR (K=1) | **CLUE** (K=1) | OTR (K=10) | **CLUE** (K=10) |
|---|---|---|---|---|---|
| umaze | 73.7 ± 7.6 | 71.4 ± 8.5 | 75.4 ± 6.1 | 75.1 ± 8.3 | 82.5 ± 5.1 |
| umaze-diverse | 21.6 ± 9.8 | 33.0 ± 8.5 | 45.4 ± 10.4 | 30.8 ± 13.5[*] | 58.6 ± 9.5[*] |
| medium-play | 23.0 ± 8.9 | 38.7 ± 11.1 | 30.5 ± 13.9 | 37.3 ± 10.0 | 36.6 ± 12.7 |
| medium-diverse | 54.9 ± 7.8 | 60.9 ± 8.7 | 64.4 ± 8.9 | 59.2 ± 9.2 | 57.8 ± 8.6 |
| large-play | 5.8 ± 3.8 | 15.0 ± 8.4 | 12.0 ± 6.5 | 13.9 ± 5.8 | 29.4 ± 8.4 |
| large-diverse | 7.0 ± 3.6 | 3.3 ± 3.6 | 0.9 ± 1.5 | 9.0 ± 5.9 | 9.7 ± 4.5 |
| antmaze-v2 total | 186.0 | 222.3 | **228.6** | 225.3 | **274.6** |

[*] Only two successful trajectories are in the chosen sub-dataset and the results belong to K=2.

**Varying the number of expert trajectories.** Using 10% of D4RL data, we vary the number of expert trajectories for sparse-reward offline RL settings in Table 5. We compare our method with baseline methods (IQL and OTR) when only one expert trajectory is selected. For comparison, we train IQL over the naive sparse-reward D4RL data and train OTR over the relabeled D4RL dataset (using optimal transport to compute intrinsic rewards and employing IQL to learn offline RL policy). We can find that in 7 out of 12 AntMaze tasks across, our CLUE outperforms the baseline OTR. Meanwhile, compared to naive IQL (with sparse rewards), our CLUE implementation generally outperforms better than IQL. This means that with only a single expert trajectory, we can completely replace the *sparse rewards* with our intrinsic reward in offline RL tasks, which can even achieve higher performance in such a data-scarce scenario (10% of D4RL data).

**Varying the value of the temperature factor in intrinsic rewards.** In Tables 6 and 7, we present the results on AntMaze tasks when we vary the value of the temperature factor $c$ in intrinsic rewards. We can find that CLUE can generally achieve a robust performance across a range of temperature factors. In Figure 7, we further analyze our intrinsic reward distribution following OTR. We can find that CLUE's reward prediction shows a stronger correlation with the ground-truth rewards from the dataset, which can be served as a good reward proxy for downstream offline RL algorithms.

Table 6: Normalized scores (mean) when varying the temperature factor $c$ with a single expert trajectory (K=1).

| | $c=1$ | $c=2$ | $c=3$ | $c=4$ | $c=5$ | $c=6$ | $c=7$ | $c=8$ | $c=9$ | $c=10$ |
|---|---|---|---|---|---|---|---|---|---|---|
| umaze | 89.4 | 89.96 | 91.84 | 90.88 | 91.96 | 92.12 | 91.68 | 90.72 | 90.92 | 91.2 |
| umaze-diverse | 43.08 | 46.76 | 43.16 | 43.76 | 42.36 | 56.72 | 52.6 | 59.04 | 66.48 | 68 |
| medium-play | 60.4 | 63.2 | 65.2 | 68.92 | 68.04 | 75.32 | 71.76 | 74.12 | 72.2 | 73.64 |
| medium-diverse | 57.8 | 63.28 | 63.24 | 62.04 | 66.04 | 70.12 | 73 | 74.56 | 69.4 | 72.92 |
| large-play | 34.16 | 44.84 | 46.88 | 50.68 | 52.72 | 53.08 | 53.64 | 55.2 | 53.52 | 55.8 |
| large-diverse | 27.04 | 33.96 | 43.16 | 46.8 | 44.88 | 47.44 | 47.44 | 49.92 | 47.28 | 47.11 |

Table 7: Normalized scores (mean) when varying the temperature factor $c$ with 10 expert trajectories (K=10).

| | $c=1$ | $c=2$ | $c=3$ | $c=4$ | $c=5$ | $c=6$ | $c=7$ | $c=8$ | $c=9$ | $c=10$ |
|---|---|---|---|---|---|---|---|---|---|---|
| umaze | 87.88 | 90 | 91.08 | 90.96 | 91.16 | 91 | 89.92 | 89.44 | 90.72 | 91.92 |
| umaze-diverse | 45.64 | 40.32 | 41.04 | 38.8 | 39.52 | 51.64 | 51.2 | 57.11 | 69.92 | 71.68 |
| medium-play | 58.72 | 64.2 | 68.24 | 71.44 | 69.92 | 75.56 | 74.12 | 76.2 | 75.8 | 76.48 |
| medium-diverse | 60.36 | 57.04 | 62.12 | 64.24 | 63.56 | 61.44 | 62.36 | 64.64 | 65.47 | 69.2 |
| large-play | 48.24 | 45.8 | 51.56 | 48.2 | 48.4 | 52.36 | 49.91 | 50.58 | 52.28 | 51.87 |
| large-diverse | 36.32 | 46.08 | 48.64 | 50.84 | 51.16 | 52.44 | 53.6 | 50.92 | 51.4 | 53.68 |

# 8 Experimental Details

## 8.1 Hyperparameters for CVAE Implementation

We list the hyperparameters used for training CVAE models in MuJoCO locomotion, AntMaze, and Adroit tasks. The other CVAE hyperparameters are kept the same as those used in Wu et al. [63].

Table 8: Hyperparameters for training CVAE.

| | MuJoCo Locomotion | | Antmaze | | Adroit |
|---|---|---|---|---|---|
| | full-data | partial-data | full-data | partial-data | full-data |
| Hidden dim | 128 | 128 | 512 | 512 | 128 |
| Batch size | 128 | 128 | 256 | 256 | 128 |
| Numbers of iterations | $10^4$ | $10^4$ | $10^5$ | $10^5$ | $10^5$ |
| Learning rate | $10^{-4}$ | $10^{-4}$ | $10^{-3}$ | $10^{-3}$ | $10^{-4}$ |
| Weight for $\mathcal{L}_{\text{calibr}}$ | 0.1 | 0.1 | 0.8 | 0.8 | 0.1 |
| Spare-reward setting: Number of expert trajectories | 3 | 3 | 5 | 5 | 3 |

## 8.2 Hyperparameters for our IQL Implementation

The IQL hyperparameters employed in this paper are consistent with those utilized by Kostrikov et al. [53] in their offline implementation. It is important to note that IQL incorporates a procedure for rescaling rewards within the dataset, which allows for the use of the same hyperparameters across datasets that differ in quality. As CLUE generates rewards offline, we similarly apply reward scaling following the IQL methodology. For the locomotion, adroit, and ant tasks, we rescale rewards with $\frac{1000}{\text{max\_return}-\text{min\_return}}$. To regularize the policy network for the chosen sub-dataset, we similarly introduce Dropout with a rate of 0.2.

**MuJoCo locomotion and Adroit tasks.** We set the learning rate $10^{-3}$ for *hopper-medium-expert* dataset (K=10) and $3 \times 10^{-4}$ for the rest of tasks. We run IQL for 1M gradient steps and average mean returns over 10 random seeds and 10 evaluation trajectories for each seed.

**Antmaze tasks.** We set the learning rate $5 \times 10^{-4}$ for *umaze-diverse* dataset (K=1 and K=10) and $3 \times 10^{-4}$ for the rest of tasks. For *medium-play* dataset (K=1 and K=10), *medium-diverse* dataset

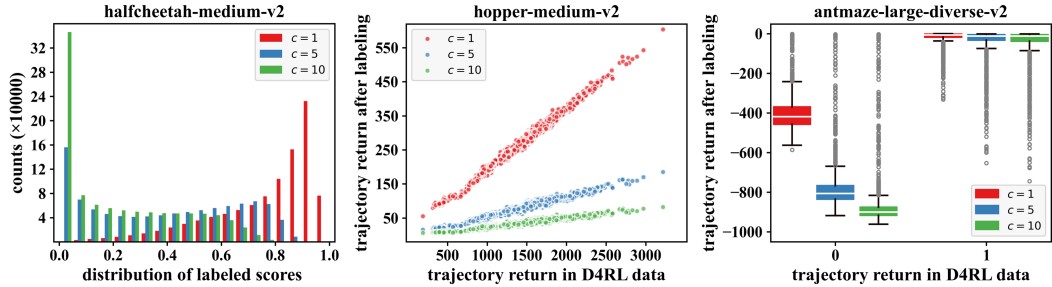

Figure 7: Qualitative comparison of the learned intrinsic rewards with different temperature factors.

(K=1), and *large-play* dataset (K=10), we set the dropout rate 0.2 to gain a better performance. We run IQL for 1M gradient steps for the full dataset and 0.3M for the partial dataset, respectively.

## 8.3 Hyperparameters in K-means

We use CLUE to learn diversity skills on *Ant-v2*, *HalfCheetah-v2*, and *Walker2d-v2*. The K-means, an unsupervised learning method, is employed to cluster the offline transitions $\{(s, a, s')\}$ from each dataset into 100 classes and take each class as a separate "expert". Specifically, we use *KMEANS* method exacted from *sklearn.cluster* API. The hyperparameters are set as follows: $\text{n\_clusters} = 100, \text{random\_state} = 1, \text{n\_init} = 1, \text{max\_iter} = 300$.

## 8.4 Offline IL Baselines

**SQIL** proposes to learn a soft Q-function where the reward labels for the expert transitions are one and the reward labels for the non-expert transitions are zero. The offline implementation of SQIL is adapted from the online SAC agent provided by Garg et al. [58], and we combine it with TD3+BC.

**IQ-Learn** advocates for directly learning a Q-function by contrasting the expert data with the data collected in the replay buffer, thus avoiding the intermediate step of reward learning. In our experiments, we used the official PyTorch implementation[5] with the recommended configuration by Garg et al. [58].

**ORIL** assumes the offline dataset is a mixture of both optimal and suboptimal data and learns a discriminator to distinguish between them. Then, the output of the discriminator is used as the reward label to optimize the offline policy toward expert behaviors. We borrowed the TD3+BC implementation reproduced by Ma et al. [60] in our experiments.

**ValueDICE** is the earliest DICE-based IL algorithm that minimizes the divergence of the state-action distribution between the learning policy and the expert data. The code used in the experiments is the official TensorFlow implementation[6] released by Kostrikov et al. [37].

**DemoDICE** proposes to optimize the policy via a state-action distribution matching objective with an extra offline regularization term. We report the performance of DemoDICE using the TensorFlow implementation[7] by Kim et al. [59], while the hyperparameters are set as same as the ones in the paper.

**SMODICE** aims to solve the problem of learning from observation and thus proposes to minimize the divergence of state distribution. Besides, Ma et al. [60] extends the choice of divergence so that

---

[5]https://github.com/Div99/IQ-Learn
[6]https://github.com/google-research/google-research/tree/master/value_dice
[7]https://github.com/KAIST-AILab/imitation-dice

the agent is more generalized. The code and configuration used in our experiments are from the official repository[8].

# 9 In What Cases Should We Expect CLUE to Help vs to Hurt?

If the distribution of the expert data is unimodal, our method can learn effectively while providing effective intrinsic rewards. On the contrary, if the distribution of the expert data is multi-modal, explicitly binding the embeddings of the expert data together would instead affect the learning of $\mathbf{z}$, thus resulting in an ineffective intrinsic reward for policy learning.

# 10 Learned Diverse Skills

To encourage diverse skills from reward-free offline data, we cluster the offline transitions into 100 classes using K-means and take each class as a separate "expert". Then, we use these expert data from different classes to label the original reward-free data and train IQL policy to learn the corresponding skills. In this section, we illustrate all the learned skills by CLUE.

---

[8]https://github.com/JasonMa2016/SMODICE

## 10.1 Learned Diverse Skills from Ant-Medium Dataset

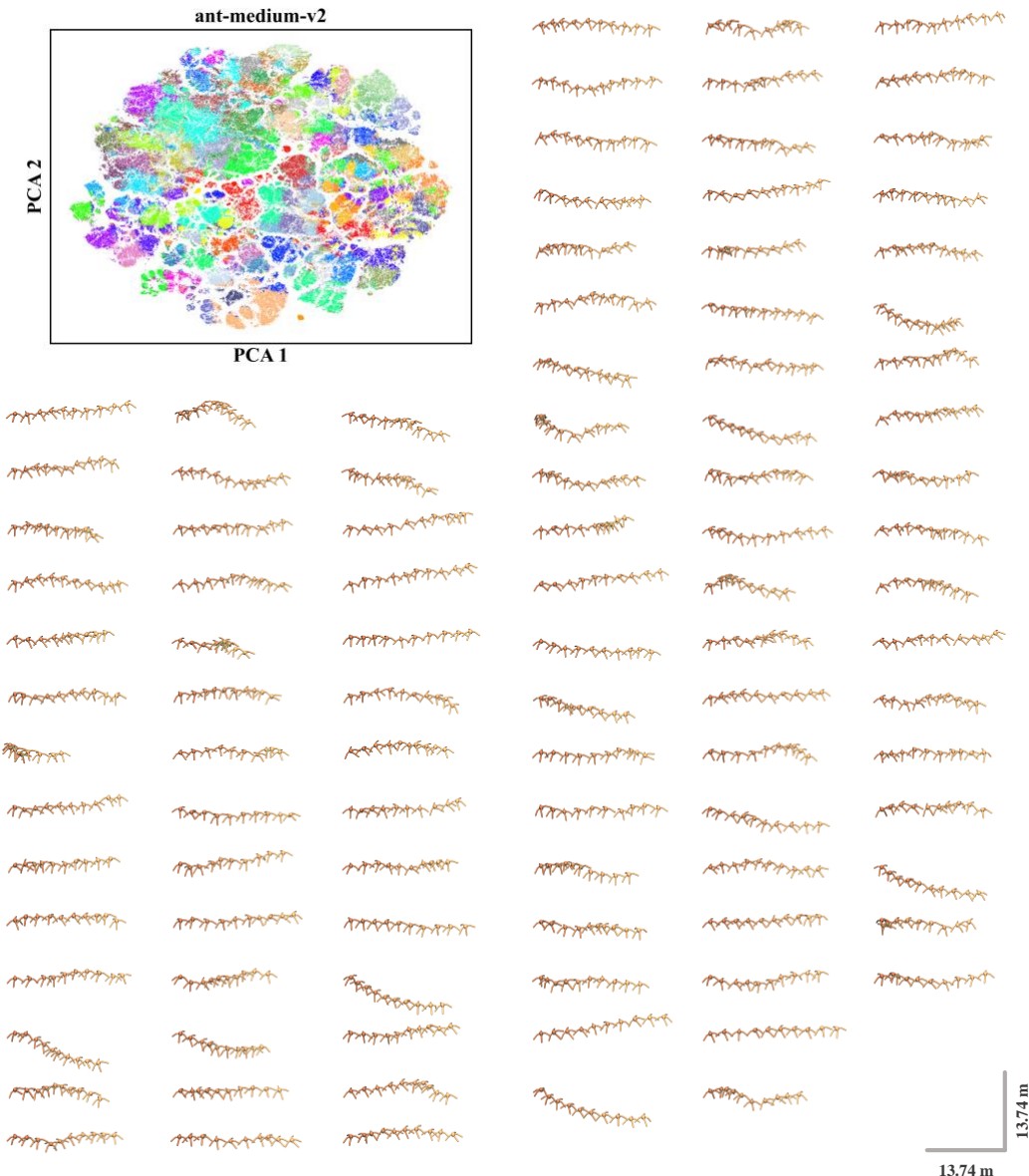

Figure 8: Visualization of unsupervised skills learned from the ant-medium dataset.

## 10.2 Learned Diverse Skills from Ant-Random Dataset

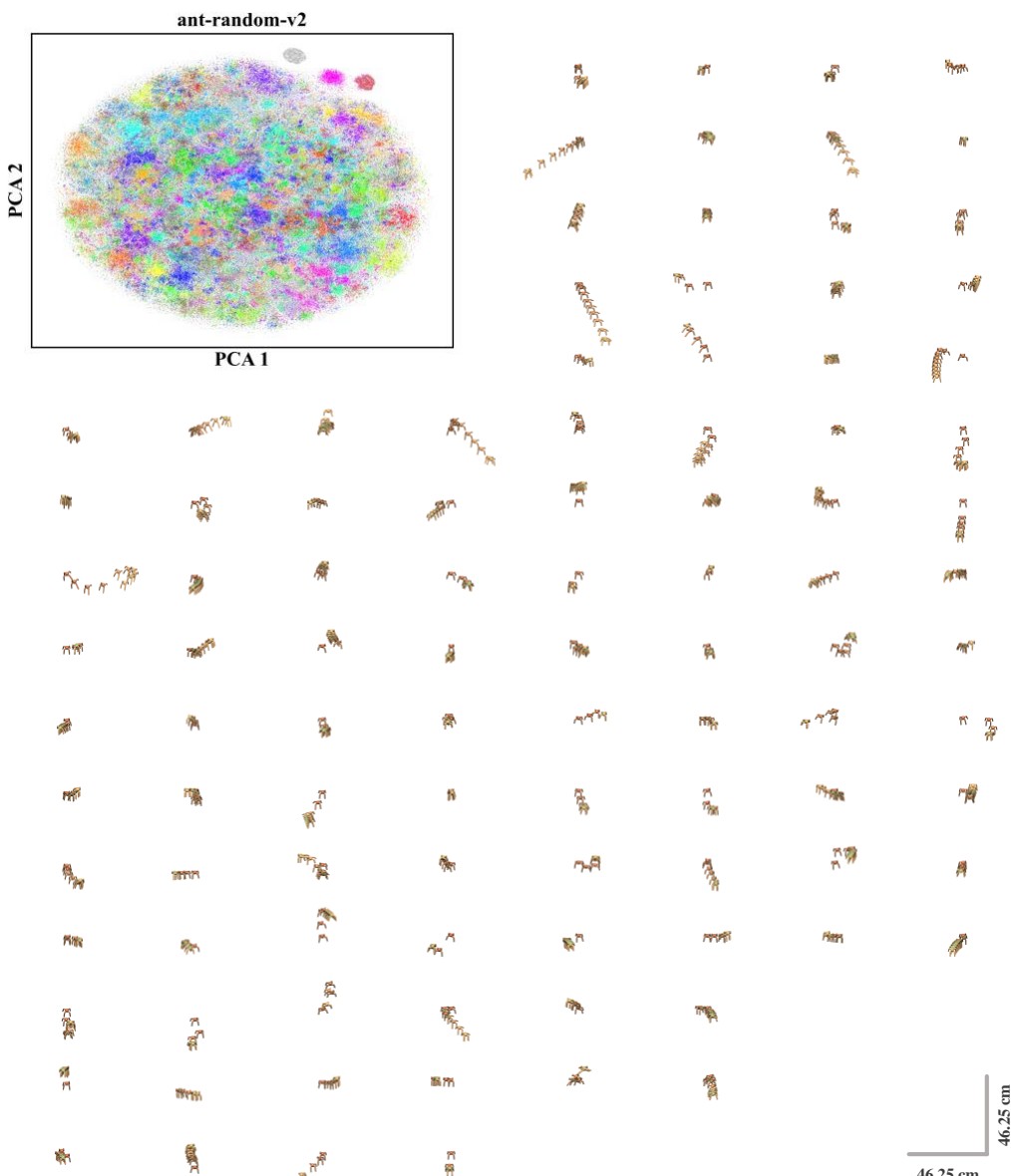

Figure 9: Visualization of unsupervised skills learned from the ant-random dataset.

## 10.3 Learned Diverse Skills from Halfcheetah-Medium Dataset

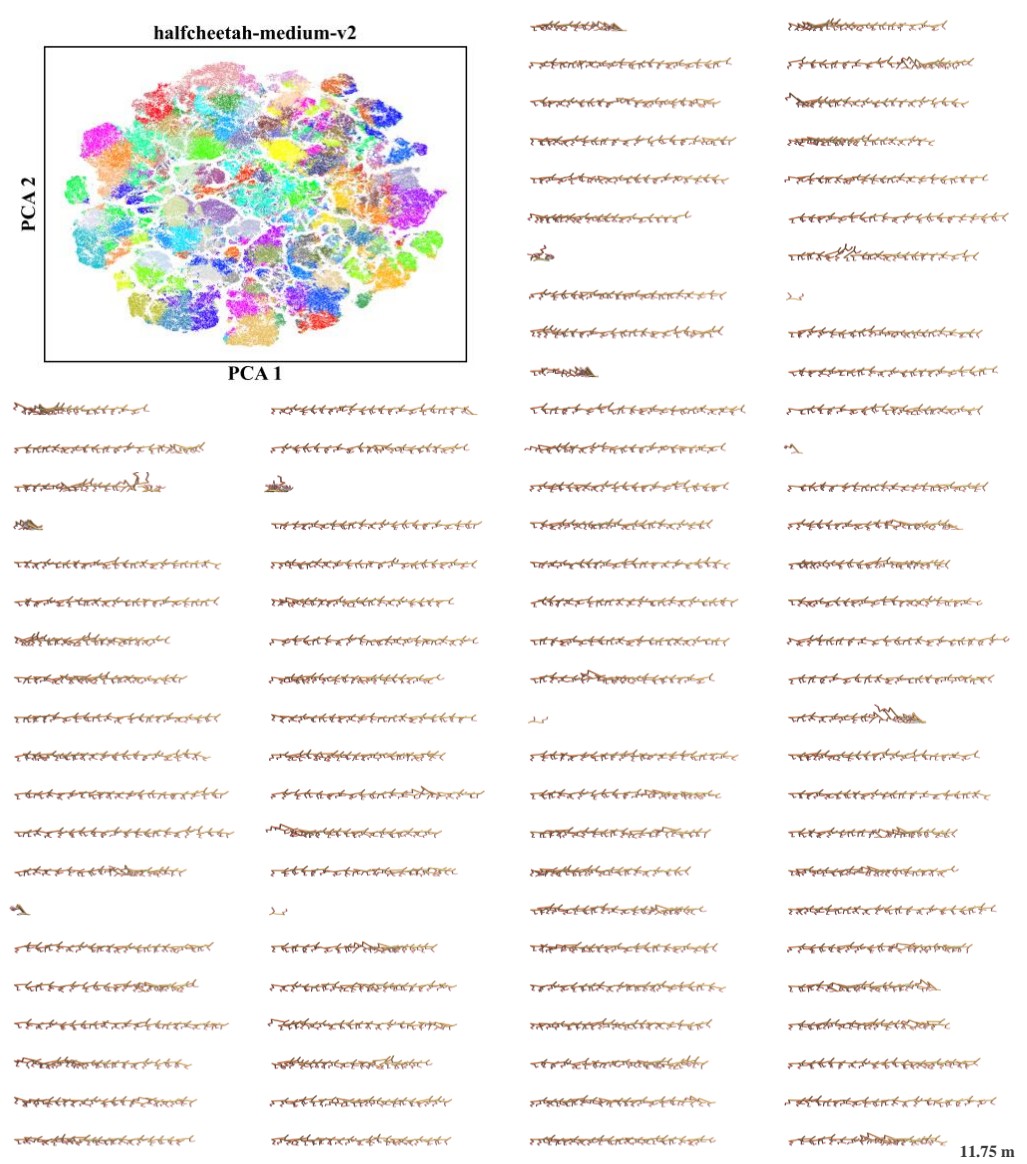

Figure 10: Visualization of unsupervised skills learned from the halfcheetah-medium dataset.

## 10.4 Learned Diverse Skills from Halfcheetah-Random Dataset

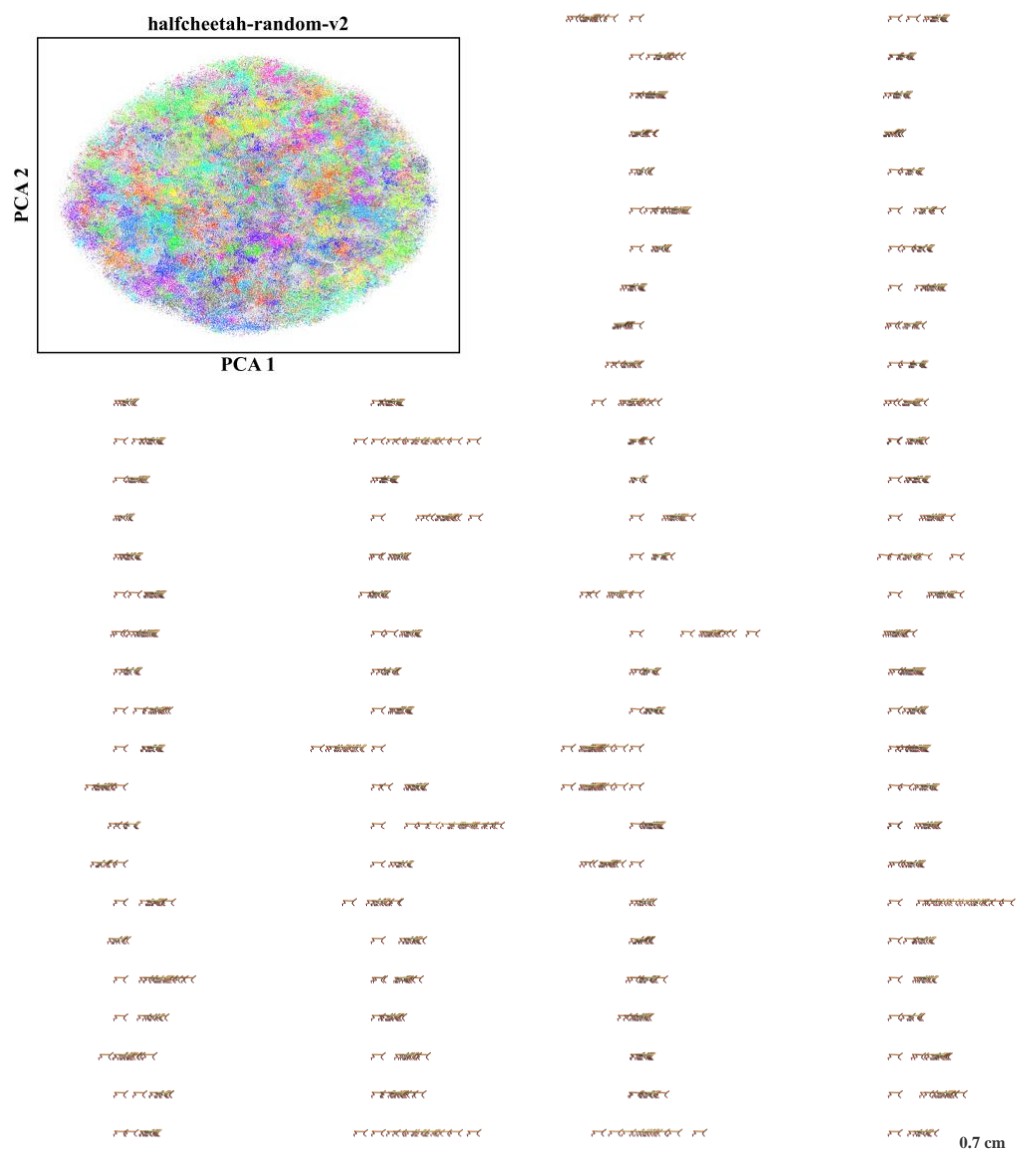

Figure 11: Visualization of unsupervised skills learned from the halfcheetah-random dataset.

## 10.5 Learned Diverse Skills from Walker2d-Medium Dataset

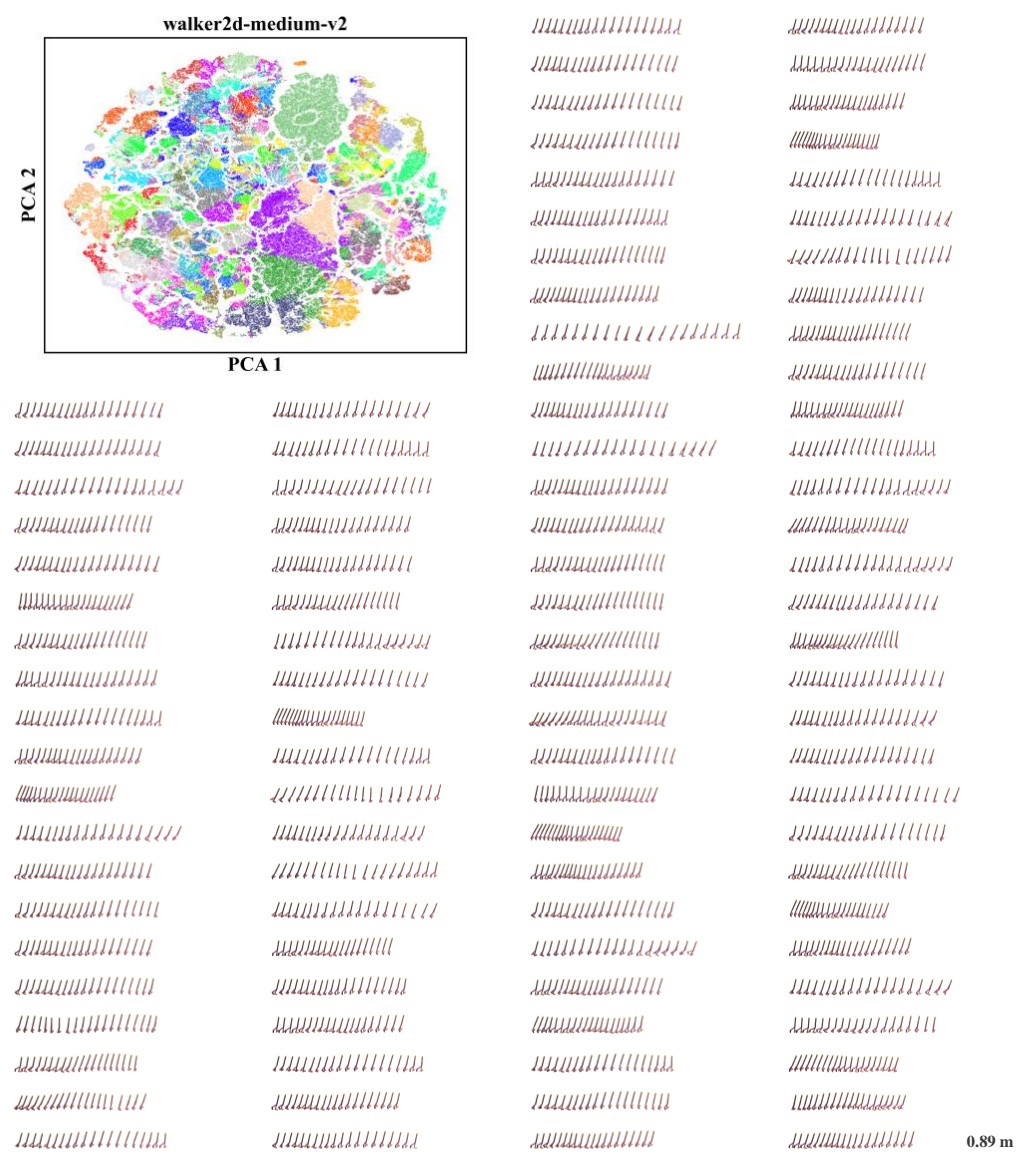

Figure 12: Visualization of unsupervised skills learned from the walker2d-medium dataset.

## 10.6    Learned Diverse Skills from Walker2d-Random Dataset

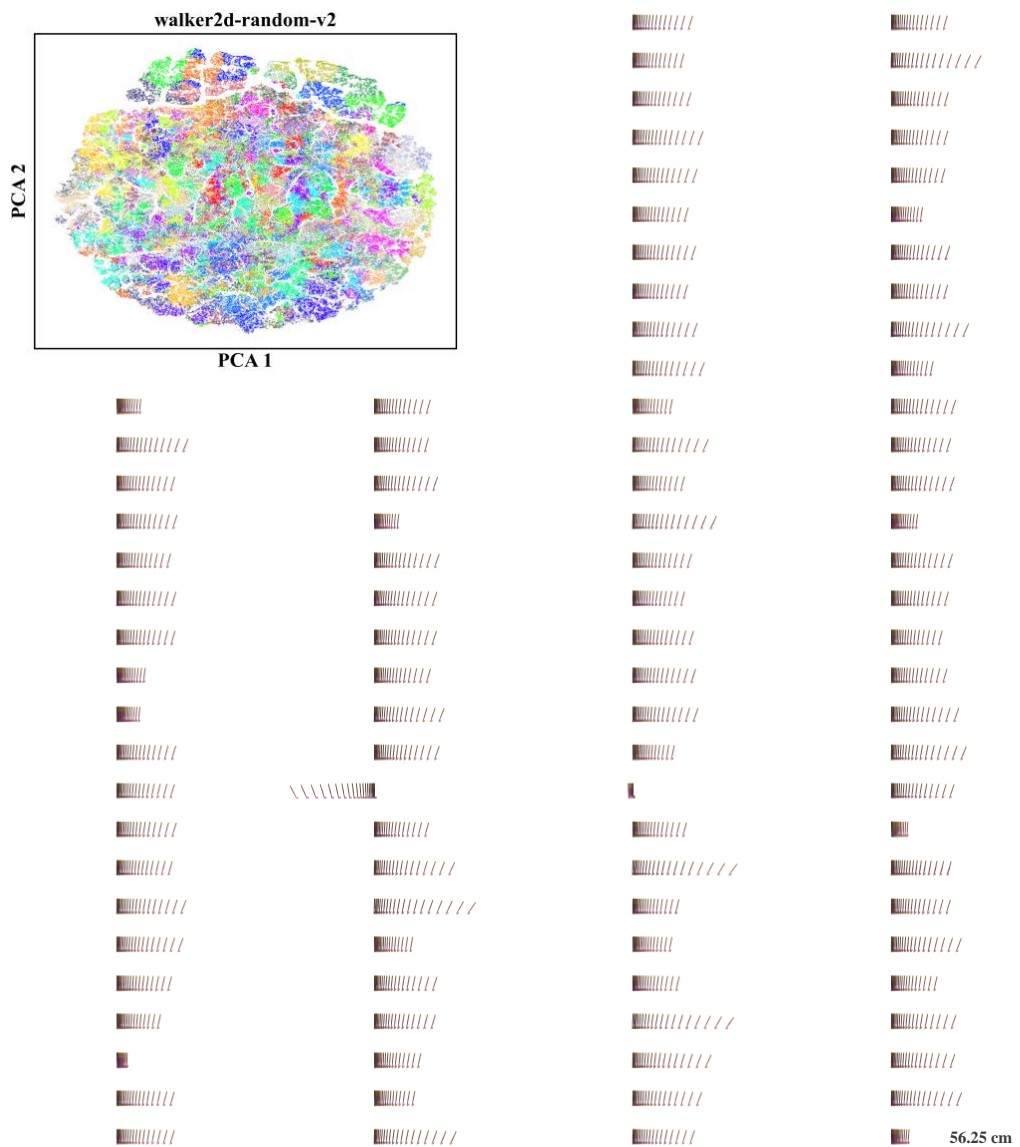

Figure 13: Visualization of unsupervised skills learned from the walker2d-random dataset.

