# OpenReview forum: "CLUE: Calibrated Latent Guidance for Offline Reinforcement Learning"
_robot-learning.org/CoRL/2023/Conference — CoRL 2023 Poster_

### Official Review · Reviewer_GEyR · 2023-07-16

**Confidence:** 5
**Originality:** Very Good
**Technical Quality:** Fair
**Clarity Of Presentation:** Good
**Impact:** 4

**Recommendation:**

Weak Reject: I recommend rejecting the paper, but will not argue for my recommendation if the majority of other reviewers have a different opinion.

**Review:**

Overall, I think the core idea in the paper is neat. I think there is definitely something to the core concept of using density estimation and latent variable models to provide auxiliary rewards or reward shaping for unlabeled data or sparse reward tasks.

On the other hand, the paper is not well motivated from a technical perspective (see more detailed comments below), and the theoretical treatment of the core concept in the work could be improved significantly.

The experimental results are thorough. While the results are not overwhelmingly strong, they do suggest that the method works, and the method is compared to a significant number of logical baselines.

It's not completely clear if the work is a good fit for CoRL, in that it's a core RL algorithms paper without robotic validation.

Specific comments:

First, regarding the technical approach:

Equation (6) is a bit puzzling for me. It states that the intrinsic reward is the exponentiated squared distance (looks like an RBF kernel) to z_e, which is defined as the expected *probability* q(z|s,a) (so the "type" of z_e is a probability?). I assume this is a typo, and in fact the authors meant to define z_e as the expected value of z sampled from the posterior q(z|s,a), averaged over the dataset. But this is still a perplexing choice: there is no reason why the average value of z should have any particular meaning -- for example, the expert data could fall into two clusters on opposite sides of the latent space, in which case the average z might be close to zero, not reflecting much information about where the expert behavior tends to lie in the latent space. Perhaps the authors meant to use a reward that roughly estimates how close z(s,a) is to expert data (e.g., something like log E_{(s',a') ~ D^e)}[q(z(s,a)|s',a')]), but the current definition doesn't really make much sense to me.

The authors instead deal with this issue by adding the somewhat ad-hoc Eq (7) objective term to cause expert embeddings to clump together. Maybe this corresponds to some well-defined probabilistic model (and I can imagine such an explanation might exist and might even provide a better motivation for Eq (6)), but as stated this seems like a rather mysterious and ad hoc decision.

I do think there is probably a much nicer way to motivate the method from the perspective of variational inference, which would provide some better justification for why we should reward the agent for transitions that resemble expert transitions, but as written now the method comes across as mostly a heuristic.

I think this is the biggest issue with the work right now.

Other (more minor) issues:

- The sparse reward offline RL results (which are closest to widely studied benchmark tasks in prior work) show some improvement, but not a very large amount. I don't think this is a major weakness, as the results are good enough, but not overwhelmingly so. I don't think this is a big point against the paper, since it still validates the core scientific idea, but I do wonder if a more principled treatment of the technical method could lead to better results.

- The discussion of intrinsic motivation in the introduction and related work is kind of misleading in my opinion. Intrinsic motivation and intrinsic rewards in online RL are typically used to drive exploration, whereas the use of the reward in this work more closely resembles reward shaping.

- There are some obvious ways of using the expert data that are not compared to, such as naively adding a BC term on the expert data, or something like this paper: https://arxiv.org/abs/2110.14770


**Quality Of The Limitations Section:**

Limitations are addressed clearly

**Questions For Rebuttal:**

The main weakness with the paper is the lack of any theoretical motivation for the proposed reward term. I would consider raising my score if the authors either provide a more complete justification of their approach, or else rederive the approach to be more principled (which may require rerunning the experiments).

Questions that would be good to answer with any such derivation include:

1. Does the method converge to the optimal policy with these reward terms under at least strong assumptions (e.g., tabular, perfect LVM, etc.)? Such proofs are easy to derive for conventional intrinsic motivation in online RL for example, or unbiased reward shaping (e.g., as in the Ng '00 paper). Could something like this be shown for the proposed method or a more principled variant of it?

2. Why is Eq (6) or Eq (7) reasonable?

3. In what cases should we expect this method to help vs to hurt?

I would raise my score if the authors convincingly address these issues.

**Robotics Focus:**

Relevant but unlikely to deploy to hardware in near future

**Summary Of Paper:**

The paper proposes to use a latent variable model to provide auxiliary rewards for offline RL. The method trains a latent variable model on both expert and non-expert data, computes the average latent variable value for expert data, and assigns rewards for other transitions based on the similarity of the inferred latent variable to this average value from expert data. The method is then applied to sparse reward tasks, imitation-style tasks (with demonstration and non-demonstration data), and clustering-based unsupervised RL, with comparisons to other offline RL and offline imitation learning methods.

**Summary Of Recommendation:**

The paper has an interesting idea, and I would strongly encourage the authors to develop this idea into a great paper.

As is however, the technical motivation for this idea is lacking, and the method comes across as ad hoc.

The relevance to robotics is also somewhat weak, as this is purely an RL algorithms paper. The algorithms in question *are* highly relevant for robotics, but the paper could well be submitted to a machine learning venue instead.

---

### Official Review · Reviewer_pVh3 · 2023-07-18

**Confidence:** 4
**Originality:** Very Good
**Technical Quality:** Very Good
**Clarity Of Presentation:** Good
**Impact:** 4

**Recommendation:**

Strong Accept: I recommend accepting the paper and will argue for my recommendation even if other reviewers hold a different opinion.

**Review:**

Overall, CLUE is a strong method for labeling intrinsic rewards, which improves offline RL performance while being able to specify tasks in the form of expert demonstrations. The major strengths are three fold. 1) It nicely leverages the auxiliary task of predicting conditional action distributions, so that the embedding space distance naturally provides a dense intrinsic reward based on similarity to expert demonstrations. The use of CVAE makes the learning easier than adversarial training. 2) Because CLUE can take any data as expert data, the user can repurpose it for different problems, including sparse reward offline RL and offline IL. 3) CLUE can be plugged in to any offline RL algorithm that can take in data with dense reward signals. These strengths are well-demonstrated throughout the paper. The figures and the tables are well-presented and make the paper easy to follow.

On the other hand, the paper has some weaknesses. 1) Unsupervised offline RL is based on clustering at the transition-level, which leaves out the possibility to cluster long-term behaviors that take more than one transition to complete (for instance, a cyclic behavior that can take 5 transitions to form a gait for the walker). 2) In the experiments section, the OTR baseline is introduced without enough details. I was unable to find the citation that corresponds to it. The authors should clarify the differences between OTR and CLUE, preferably in the Related Work section.

**Quality Of The Limitations Section:**

Limitations are addressed clearly

**Questions For Rebuttal:**

* The practical use of unsupervised offline RL remains a bit unclear. How would you determine which specific skill to use, given a task specification?
* In addition to performing calibration on the expert data, the authors could have also added another loss term on the non-expert data to further separate those embeddings from the expert ones (for instance by encouraging larger stdev for non-expert data). Did you consider doing that?

**Robotics Focus:**

Highly relevant to robotics but no hardware experiments

**Summary Of Paper:**

This paper proposes CLUE, a learning framework for labeling intrinsic reward functions to offline transition data. The key idea is to perform an auxiliary task of learning conditional action distributions on all the offline data, and leverage the learned latent embedding space to compute the intrinsic reward. Specifically, the reward is computed as the negative distance between the latent embeddings of the expert data and the query sample. It assigns a larger reward value to a state-action pair that is more similar to the expert data, and vice versa. The paper applies CLUE with an offline RL algorithm to perform sparse-reward offline RL, offline IL, and unsupervised offline RL. The learned policies outperform state-of-the-art baselines in sparse-reward offline RL and offline IL tasks.

**Summary Of Recommendation:**

Despite some weaknesses, the paper makes a major novel contribution to automatic reward design for offline RL and offline IL. The potential impact of the work is quite high, given the relevance of those problems to robotics as well as the strong empirical results presented in the paper. Therefore, acceptance is recommended.

---

### Official Review · Reviewer_CYF2 · 2023-07-23

**Confidence:** 2
**Originality:** Good
**Technical Quality:** Very Good
**Clarity Of Presentation:** Fair
**Impact:** 4

**Recommendation:**

Strong Accept: I recommend accepting the paper and will argue for my recommendation even if other reviewers hold a different opinion.

**Review:**

Clarity: The method is described clearly in section 4. I found the introduction to be too all-encompassing, which made it difficult to understand what the paper actually does. Specifically, paragraphs 2 and 3 cover everything from intrinsic rewards to augmented reward terms in offline RL. Although it was nice to see the connection to multiple problem statements within RL, I think it would make the paper stronger if the motivation for CLUE was described more succinctly and directly.

Originality: The method is logical and novel. I appreciated how the authors demonstrated how the method could be applied in multiple problem statements (e.g., by clustering the trajectories when no expert data is available).

Significance: Leveraging a cVAE could provide a new unifying framework for multiple offline sequential learning problem statements, which feels significant.

Quality: Claims are verified against a reasonable collection of relevant baselines (IQL, OTR).

**Quality Of The Limitations Section:**

Limitations are addressed clearly

**Questions For Rebuttal:**

In equation (1) why is the KL term negative. Following from equation (1) in Kingma et. al. (2022), shouldn't it be: $\log p_\theta(x) = KL(q_\phi(z|x)||p_\theta(z|x)) + \mathbb{E}_{q_\phi(z|x)}[- \log_q(z|x)+ \log_p(x,z)]$?

I find the presentation of the 3 problem statements confusing. E.g., this may be a bit pedantic but isn't offline-RL without labels just an IL problem with messy data? Also in most of these cases it seems reasonable to assume that we have access to some offline reward function (even if it's quite naive, such as labeling each trajectory with a sparse reward of 1 at the terminal state). Could simple reward functions such as that be a strong baseline for these tasks?

D. P. Kingma and M. Welling. Auto-encoding variational bayes, 2022.

**Robotics Focus:**

Relevant but unlikely to deploy to hardware in near future

**Summary Of Paper:**

The paper studies a modification of the offline RL problem statement in which there is a small amount of expert data that can be used to create an intrinsic reward function. The intrinsic reward function is the distance of non-expert data from expert data in the embedding space of a cVAE. The authors apply this approach to improve performance on sparse-reward tasks, offline imitation learning tasks with some expert trajectories labeled, and offline RL assuming no expert trajectories. The datasets for each environment come from D4RL.

**Summary Of Recommendation:**

I recommend accepting this paper because the idea of deriving methods for multiple offline RL problem statements by aligning latents with expert trajectories is powerful and the claims are well evaluated.

---

### Official Review · Reviewer_ashV · 2023-08-01

**Confidence:** 3
**Originality:** Good
**Technical Quality:** Good
**Clarity Of Presentation:** Very Good
**Impact:** 3

**Recommendation:**

Weak Accept: I recommend accepting the paper, but will not argue for my recommendation if the majority of other reviewers have a different opinion.

**Review:**

**Strength:**

- The paper is generally well written and has a good amount of experiments.
- The proposed calibration regularisation term is intuitive, novel, easy to implement by itself and should introduce minimal training time overhead.
- Strong empirical results on the 3 instantiations of expert data in offline RL settings.
- Experiment results on offline IL tasks versus baseline algorithms (Table 4) are particularly impressive.
- Supplemented code is appreciated.


**Weaknesses:**

- The justification for using CVAE is not well motivated.
- The proposed calibration regularisation needs to be applied with CVAE together.
- Since the paper’s claim is that the proposed regularisation term can be combined with any offline RL algorithms, comparisons using other base algorithms would be preferable. Is there a particular reason that IQL was chosen as the base algorithm?
- The temperature hyperparameter $c$ used in various experiments cannot be found in the texts.


 **Minor notes:**

- How to apply the proposed calibration regularisation term is not clearly written (i.e., where the loss should be applied). It can be inferred from the texts but can be written more clearly.
- typo in line 105, I think $\mu_0$ should be $p_0$?
- I’m a little bit puzzled by the supplemented code; for CVAE, it seems the code is implemented as a beta-CVAE (with beta=0.5) whereas there is no mention of this in the text.
- The hyperparameters of OTR and some other baselines (SQIL and ORIL) were not mentioned.

**Quality Of The Limitations Section:**

Limitations are addressed clearly

**Questions For Rebuttal:**

1. In Appendix, it seems that the choice of temperature hyperparameter $c$ could impact the final performance in a non-insignificant way (Table 1 in the main text; Table 2 and Table 3 in Appendix). On a closer look at the 3 tables, it seems that the result from Table 1 in the main text is an ensemble of results trained under different hyperparameters $c$ (i.e., *medium-play* is from $c=6$ while *medium-diverse* is from $c=8$, etc, from table 2 in Appendix). Could the authors clarify? Is this also the case for other results presented?

2. For the 2nd instantiation – The result in Tables 3 and 4 for locomotion tasks hopper/walker2d/half-cheetah with K=1 seems impressive. How about more complicated tasks such as Adroit or Antmaze to further understand the results compared to OTR? Additionally, OTR hyperparameters should be included for completeness.

3. For the 3rd instantiation - What are the baselines for this instantiation of expert data usage? Is there a quantitative way to better understand the results?

4. It is unclear to me how to interpret Figure 2 correctly. Is it generated from CVAE/CLUE trained on the whole expert data $\mathcal{D^e}$ vs trained on whole non-expert data $\mathcal{D}$?
    1. What do the authors think about the result where the CVAE std centre seems to be around $\sigma = 0.9$ and, after calibration, the std centre seems to shrink towards $\sigma = 0.96$? This is also the case for non-expert data.
    2. How good must the expert demonstration be so that the latent embeddings will shrink towards $\sigma = 1$?

5. Related to question 4, can the method still perform well if, instead of using the "optimal" expert demonstration in the dataset, let's say uses the demonstrations in the 90th percentile? How sensible is the proposed method to expert demonstrations compared to OTR? (i.e., how much does a chosen demonstration affect the method?)

**Robotics Focus:**

Relevant but unlikely to deploy to hardware in near future

**Summary Of Paper:**

The paper proposes a new approach to learning from very few expert demonstrations. Their proposed method estimates intrinsic rewards that are more in line with extrinsic task-specific rewards obtained from expert data. The key idea is to add a calibration regularisation term to CVAE to estimate the intrinsic rewards. This calibration reduces the variances of estimated mean and std in CVAE for expert data. The authors demonstrate that the proposed calibration yields good results in 3 settings of expert data usage in offline RL.

**Summary Of Recommendation:**

This paper proposes an effective and intuitive approach to tackle the problem of learning from very few expert demonstrations. Overall the writing is clear and demonstrates strong empirical results. I am leaning towards accepting this paper, as long as the concerns listed above are addressed appropriately.

---

### Decision · Program_Chairs · 2023-08-30

**Decision:**

Accept (Poster)

**Comment:**

This paper presents a framework that leverages a few expert data and obtains intrinsic rewards that guide an offline RL agent. The effectiveness of the proposed method was evaluated with sparse-reward offline RL tasks, offline imitation learning tasks, and unsupervised offline RL tasks.

The initial evaluation was mostly positive, and the authors addressed most concerns raised by reviewers in the initial review.
While Reviewer GEyR pointed out that the derivation of the proposed algorithm could be more formal, all reviewers agree that the idea in this paper is interesting and novel.

Based on the assessment from reviewers, AE recommends the acceptance of the paper.
AE strongly encourages the authors to carefully incorporate the points raised by reviewers before submitting the final version. Especially, please check the notation and equations carefully once again.